# Investigation of Wood Flour Size, Aspect Ratios, and Injection Molding Temperature on Mechanical Properties of Wood Flour/Polyethylene Composites

**DOI:** 10.3390/ma14123406

**Published:** 2021-06-20

**Authors:** Mohammad E. Golmakani, Tomasz Wiczenbach, Mohammad Malikan, Reza Aliakbari, Victor A. Eremeyev

**Affiliations:** 1Department of Mechanical Engineering, Mashhad Branch, Islamic Azad University, Mashhad 9187144123, Iran; m.e.golmakani@mshdiau.ac.ir (M.E.G.); rezaaliakbari292@gmail.com (R.A.); 2Department of Mechanics of Materials and Structures, Gdansk University of Technology, 80-233 Gdansk, Poland; mohammad.malikan@pg.edu.pl (M.M.); or eremeyev.victor@gmail.com (V.A.E.); 3Department of Civil and Environmental Engineering and Architecture, Università degli Studi di Cagliari, Via Marengo, 2, 09123 Cagliari, Italy

**Keywords:** wood flour size, polyethylene, tensile strength, tensile modulus, flexural strength, flexural modulus, impact energy, injection molding temperature

## Abstract

In the present research, wood flour reinforced polyethylene polymer composites with a coupling agent were prepared by injection molding. The effects of wood flour size, aspect ratios, and mold injection temperature on the composites’ mechanical properties were investigated. For the preparation of the polymer composites, five different formulations were created. The mechanical properties including tensile strength and the modulus, flexural strength and the modulus, and impact energy were measured. To investigate the changes in the properties resulting from different compositions, mechanical static and impact testing was performed. The obtained results indicate that by reducing the flour size, the tensile strength and modulus, flexural strength, and impact energy were reduced. In contrast, the flexural modulus increased. Furthermore, with the increment of injection molding temperature, the tensile strength and the modulus and the impact energy of the specimens were reduced. On the other hand, the flexural strength and the modulus increased. Thus, an optimized amount of injection molding temperature can provide improvements in the mechanical properties of the composite.

## 1. Introduction

Composites made from wood fibers and polymers have become commercialized and have grown rapidly in various wood replacement applications. Besides being used as construction products, wood–plastic composites (WPCs) are used in many industries such as automotive and unique industrial structures. A wide range of cellulose and lignocellulosic polymers and fillers are used in the manufacture of these composites. Hence, the broad interest in natural origin fillers is an alternative to traditional, commonly used plastic fillers [1,2] that are relatively cheap, renewable, have good mechanical properties, low processing shrinkage, and are significantly water-resistant [3,4]. In addition, they resemble wood, which affects their aesthetic values. Depending on the aspect ratio of the wood filler, products are obtained with properties similar to polymer materials or wood products [5].

For the production of WPCs, thermoplastic polymers based on polyolefins (polypropylene PP, polyethylene PE) or polyvinyl chloride (PVC) are used. Waste wood can be used as a filler (shavings, sawdust, dust, lumber, and other fractions). WPCs have many advantages from wood and other wood-based materials (plywood, chipboards, OSB, MDF, etc.), resulting from significant resistance to weather conditions [6,7]. It makes these composites willingly used as a substitute for wood to produce terraces, platforms, or building facades [8,9].

However, the increasing aspect ratio of wood in the composite significantly affects the deterioration of the mechanical properties of WPCs [10]. This is mainly due to the low interfacial adhesion between the hydrophobic polymer matrix and the hydrophilic wood filler.

Mahanty et al. studied the effect of fiber length, adapter concentration, adapter treatment time, and fiber weight percentage on the composite’s mechanical properties. It was concluded that with increasing fiber length, treatment time, adaptive amount, and weight percentage of fibers, the adhesion between the two phases increases flexural strength and water absorption decreases [11].

Chen et al. investigated the effect of wood particle size and heavy polyethylene mixing ratios on the composites’ properties. He reported that larger structures with better dimensions had better strength [12].

Magnolt et al. investigated the effect of dimension length on polyethylene composites and wood fibers. He concluded that the tensile and flexural strength and the tensile and flexural modulus increased with increasing fiber dimensions [13].

Magnolt et al. investigated the effects of fabrication methods and fiber dimensions on the structure of plastic–wood composites and their mechanical properties. In this research, the pulp fiber filler and its matrix were heavy polyethylene. The results showed that reducing the fibers’ dimensions (reducing the ratio of length to diameter), water absorption, thickness swelling, strength, tensile modulus, strength, and flexural modulus decreased [14].

Huang et al. investigated the effect of wood flour size on the mechanical properties of the plastic–wood composites. Their research showed that increasing the aspect ratio of the tensile modulus of the composites increased by 28.4%, and with smaller dimensions, the impact energy of composites increased by 35.5% [15].

Much research has been done on the polymer composites’ physical and mechanical properties based on wood particles, most of which have focused on the mechanical properties and hydrophilic properties of wood, and the hydrophobic properties of plastic matrices create better compatibility between these two materials [16,17,18]. However, little research has been conducted on the thermal properties and heat effect on mechanical properties focusing on the thermal parameters for WPCs. One of the main problems of WPC is the weak connection between polar and hydrophilic wood fibers with hydrophobic and nonpolar polymers, which strongly affects the mechanical properties of the final product [19].

Coupling materials such as maleic anhydride are usually used to create a better bond between the polymer material and the wood flour [20]. However, many variables affect the interconnection between particles in the polymer network, such as fillers, binders, and process aids during production. In addition, process variables such as mixing temperature and injection molding speed also affect the final properties of the product [21].

According to the above, this study was conducted to investigate the effect of wood flour and its aspect ratio on the mechanical properties of the wood flour/polyethylene composites. Additionally, injection molding temperatures on the mechanical properties of wood flour/polyethylene composites were investigated.

## 2. Materials and Methods

### 2.1. Materials

In this study, three wood flour levels were considered to investigate the mesh size contribution to the mechanical properties. The classification was performed to uniformize the particle size and reach the desired size with a vibrating sieve. Three different reinforcement sizes were considered: coarse-grained, medium grain, and fine with approximately +30/−40, +70/−80, and +100/−120, respectively. For the investigation of the injection molding temperature effect on the mechanical properties, only the +100/−120 mesh size was considered. The grains were placed in an oven at 100 °C for 24 h until their humidity reached 2%.

Heavy polyethylene by Arak Petrochemical Company (Tehran, Iran) with code 5218 and melt flow index 18 g/10 min and density 0.959 g/cm^3^ was used as a matrix.

Furthermore, to create compatibility between the reinforcement and matrix, maleic anhydride-grafted-polyethylene (MAPE) produced by Krangin Company (Karaj, Iran) was used. MAPE’s main specifications are a melt flow index of 7 g/10 min and density of 0.965 gr/cm^3^. This supplement was used at a constant level of 3 wt.%. 

### 2.2. Wood–Plastic Composite Specimens Preparation

The mixing process was done using a HBI System 90 internal mixer created by the Haake Buchler American Company (Hainesport, NJ, USA). Mixing at the speed of 60 rpm and a temperature of 170 °C was performed. The total mixing time before making constant torque was 10 min. First, heavy polyethylene was poured into the device, and afterward, melted for 2 min; when gaining constant torque, MAPE was added. After 5 min, the wood flour was added.

After hardening and cooling, the material was granulated utilizing a Wieser WG-Ls 200 semi-industrial shredder (Wieser Company, Hamburg, Germany). Table 1 shows the compounds and injection molding temperature for each composition of the specimens.

Each composition’s grains were dried in a dryer at 80 °C for 24 h. An MPC-40 semi-industrial injection molding machine (Aslanian Machine, Tehran, Iran) at three different temperature levels of 170, 185, and 200 °C was used. An injection pressure of 80 bar was considered. Specimens were made for tensile, flexural, and impact energy tests according to ASTM D638-10 [22], ASTM D790-10 [23], and ASTM D256-10 [24] standards, respectively. Before the mechanical tests, the samples were exposed to climatic humidity (temperature 20 ± 2 °C and relative humidity 65%) for two weeks. Figure 1a–c presents the tensile, bending, and impact manufactured test specimens, respectively. Additionally, Figure 2a–c presents the dimensions of the specimens for the tensile, bending, and impact test, respectively.

### 2.3. Mechanical Testing

ASTM D638-10, ASTM D790-10, and ASTM D256-10 standards were used for the tensile, flexural, and impact energy tests, respectively. The mechanical tests of the samples were performed on an Instron 1186 (Instron, Norwood, MA, USA) testing machine at 20 ± 2 °C room temperature. The impact test was performed on a Zwick Model 5102 (Zwick GmbH & Co. KG, Ulm, Germany) at the same room temperature and humidity. Bending and tensile tests were performed with an initial 500 N load cell force. The crosshead speed was considered at a constant level of 2 mm/min. All mechanical testing for each specimen composition was performed a minimum of five times. Following the force–displacement graph obtained from the experimental test, the computations were done to achieve Young’s and flexural moduli and tensile and flexural strength. The mean values from the calculations were considered. Additionally, according to the obtained curves from the impact test, the impact energy was computed.

### 2.4. Filler Dimension Analysis

One of the critical and influential factors on the properties of composites is the aspect ratio of the filler. Therefore, a light microscope (Peybord, Tehran, Iran) equipped with a sample size analyzer was used to obtain the filler aspect ratio.

### 2.5. Statistical Analysis

The SPSS 24 program (IBM Corp., Armonk, NY, USA, 24.0) was used for statistical analysis in this research. Data analysis was performed using the factorial test in a completely randomized design. Analysis of variance (ANOVA) was computed to check if mean differences were significant. Duncan’s multiple domain test was used to compare the mean values. The effect of the variable factors on the computed mechanical properties was analyzed at the confidence level of 95% (5% significance level).

## 3. Results

### 3.1. Effect of Mesh Dimensions on Mechanical Properties

#### 3.1.1. Dimensions of Flour

Table 2 shows the average length, diameter, and aspect ratio of wood flour in three categories of +30/−40, +70/−80, and +100/−120 meshes.

As shown in Table 2, the aspect ratio of wood flour increased from +30/−40 to +70/−80 with the reduction in mesh dimensions and then decreased to +100/−120 mesh.

#### 3.1.2. Experimental Test Results

The variance analysis of the effect of mesh dimensions (aspect ratio) on the flexural strength, modulus, tensile strength, and impact energy of the composites is shown in Table 3. According to Table 3, the effect of mesh dimensions on impact energy, flexural, and tensile strengths was significant but not substantial on the other studied properties. Statistical significance when less than 5% of the relationship was due to an accident. The determination of 95% is arbitrary and chosen.

Figure 3a,b presents the impact of flour size on the tensile modulus and strength, respectively. The black lines show the standard deviation. Following Duncan’s test grouping, the maximum tensile modulus was related to the specimens made of +70/−80 mesh flour (6067.3 MPa), and the minimum in group A was connected to the samples made of +100/−120 mesh flour (4655.3 MPa). According to Duncan’s test grouping, the maximum tensile strength in group A was related to the samples made of flour mesh +30/−40 (26.4 MPa) and flour +70/−80 (26.2 MPa). The minimum in group b was connected to specimens made from the +100/−120 size (21.5 MPa).

Figure 4a,b shows the effect of flour dimensions on the flexural modulus and strength, respectively. Black lines represent the one measure of standard deviation. Following the Duncan test, the maximum flexural modulus was related to the samples made of +100/−120 powder (4007.7 MPa), and the minimum in group A was connected to the specimens made of +30/−40 flour size (3795.3 MPa). According to Duncan’s test grouping, the maximum flexural strength in group A was related to samples made of +70/−80 size (49.1 MPa). The minimum in group b was connected to specimens made of +30/−40 (45.9 MPa) and +100/−120 (45.3 MPa) flour size.

Figure 5 presents the effect of wood flour size on impact energy. The highest impact energy was obtained for wood flour with dimensions of +30/−40 powder size. The impact energy was lowest for the +100/−120 mesh.

### 3.2. Effect of Injection Mold Temperature on Mechanical Properties

The variance analysis of the effect of injection mold temperature on tensile strength and the modulus, flexural strength and the modulus, and impact energy of composites are shown in Table 4. According to Table 4, the effect of injection mold temperature on tensile and impact energy was significant. On the other hand, this effect was negligible on the tensile and flexural modulus and flexural strength.

Figure 6a,b presents the effect of injection mold temperature on the tensile modulus and strength, respectively. By increasing the injection mold temperature to 185 °C, the tensile strength and modulus initially increased significantly, but a further increase to 200 °C crucially reduced the strength of the specimens. Following the Duncan test, the maximum tensile modulus was related to the samples made at the temperature of 185 °C (5471.7 MPa), and the minimum was associated with the specimens made at a temperature of 200 °C (4655.3 MPa). By increasing the injection mold temperature to 185 °C, the tensile strength initially increased significantly, but a further increase to 200 °C crucially reduced the strength of the specimens. According to Duncan’s test grouping, the maximum tensile strength in group A was related to the samples made at 185 °C (28.3 MPa). The minimum in group B was connected to the made pieces at a temperature of 200 °C (21.5 MPa).

Figure 7a,b presents the effect of injection mold temperature on the flexural modulus and strength, respectively. The maximum flexural modulus corresponds to the specimens made at the mold temperature of 185 °C (4107.7 MPa). The minimum that belonged to one group corresponding to the samples made at the injection mold temperature of 170 °C (3478.7 MPa). As the injection mold temperature increased, the flexural strength first increased, and then with a further rise in temperature, the strength decreased, which was not statistically significant. The maximum flexural strength was related to the samples made at the injection temperature of 185 °C (48.4 MPa). The minimum strength was related to the pieces made at the pressing temperature of 170 °C (43.9 MPa).

Figure 8 presents the effect of injection mold temperature on impact energy. The lowest value of impact energy was obtained at the injection mold temperature of 200 °C. The impact energy was highest at a mold temperature of 185 °C.

As the injection mold temperature increased from 170 to 185 °C, the tensile strength and modulus, flexural strength and modulus, and impact energy increased by 10.1, 11.4, 10, 18.1, and 26%, respectively. Furthermore, with an increment to 200 °C, the tensile strength and modulus, and impact energy decreased by 19.5, 5.5, and 4.2%, respectively. On the other hand, the flexural strength and modulus increased by 3 and 5.9%.

For numerical simulation and processing modeling, it is crucial to obtain the thermophysical properties by mathematical expressions of polymer-based composites. There are some other works in which this has been performed for different WPC compositions [25] and references therein.

## 4. Discussion

In general, the mechanical properties of wood-their components’ properties influence plastic composites. It should be noted that the homogeneity of wood-plastic (distribution of wood flour and its wetting) was necessary to improve the properties [26].

The highest tensile strength and modulus were obtained in composites made of wood flour with dimensions of +70/−80 mesh. The increase in tensile strength and modulus with the size of flour in mesh +70/−80 can be attributed to the rise in the length to the diameter ratio of flour. Increasing the aspect ratio of flour is likely to neutralize more stress in the polymer matrix than particles with lower viscosity [27]. As a result, the tensile strength and modulus increase. Besides, as the particles become smaller, the stress transfer between the particles may be more heterogeneous due to the lower strength and tensile modulus of +100/−120 mesh compared to +30/−40 mesh. Nourbakhsh et al. [21] stated that having more contact surfaces with the ground phase in smaller particles improved the tensile strength. They also noted that the tensile strength increased with an increase in the aspect ratio. Some researchers have also stated that the composite’s tensile strength is reduced by reducing the particle size. Caraschi and Lopes [28] noted no significant difference between mechanical strength and composite modules made of Elliott pine with wood particle dimensions. Lai et al. [29] also pointed out that the tensile strength decreased as the flour’s size decreased because of the reduction in bonding surface between the polymer and the filler. One side of the specimen is pulled in the tensile test, and the other is compressed due to the applied force. Therefore, the amount of dispersion and wetting of fibers affects this feature [30].

It was observed that as the dimensions of flour as a reinforcement became bigger, up to the +100/−120 size, the flexural modulus increased. Therefore, it is possible that reducing the aspect ratio increases the effect of filler reinforcement in composites and improves the stress distribution in the specimens, which increases the flexural modulus. Another reason is the uniform mixing of particles with a lower viscosity coefficient of the polymer due to the higher contact surface of these particles and better injection in the molding machine. Cui et al. [31], Williams [32], and Febrianto et al. [33] stated that due to the more excellent contact surface, fine-grained particles are more uniform, and the flexural modulus increases.

Moreover, as the particles become finer, the particles’ stress transfer is more homogeneous and increases the resistivity. It is also possible that as the particles become finer, the mixing between the lignocellulosic material and the polymer matrix becomes better and more uniform. On the other hand, the fine particles pass through the injection hole better. Better injection of particles can increase the resistance. These results have also been reported by many other researchers [27,31,33]. As the particle size becomes smaller, the impact energy increases.

Furthermore, it can be said that perhaps larger particles act as stress agents and create places for the initial onset of cracking, causing the composite to break more easily. Additionally, short fibers (fine particles) have a higher specific surface area due to their greater frequency. As a result, they have a more uniform distribution, and there will be more compatibility between the fibers and the underlying material [34].

The highest values of tensile and flexural strength and modulus were obtained in composites made at a temperature of 185 °C. This may be due to the greater dispersion of the filler within the polymer matrix, which causes the uniformity of the filler in the polymer matrix and increases the interaction between them.

The mechanical properties improved with better filler dispersion in the polymer field [35,36]. Improvement in the mechanical properties of the composites requires strong adhesion between the reinforcement and the polymer. The composites’ final properties crucially depend on the manufacturing conditions, such as the process and its requirements. Mixing at the right temperature to achieve the desired dispersion improves the properties of the composite [37]. The decrease in mechanical properties at 200 °C could be due to the formation of acidic chemicals such as acetic acid or formic acid resulting from the decomposition of hemicellulose [38]. These acids dissolve cellulose by breaking the long chain of cellulose into shorter chains. In addition, carbon-carbon and carbon-oxygen bonds disappear at the polymer surface with increasing temperature. This leads to the separation of the copolymer system of lignin, hemicellulose, and amorphous cellulose [39]. Cellulose chain shortening affects the resistance properties [40]. Furthermore, concerning the strength properties, the excessive increase in mold temperature reduces the strength because the strength properties are strongly dependent on the polymer matrix structure. The rise in temperature causes the degradation and decomposition of the matrix, and therefore, the mechanical properties.

## 5. Conclusions

In general, the mechanical properties of wood-plastic composites are affected by their components’ properties. The joint’s quality between the polymer mixture and the bond between the polymer matrix and the lignocellulosic material determines the process conditions.

In this study, the investigation of wood flour size on the mechanical properties of WPCs was performed. Wood flour was used in three +30/−40, +70/−80, and +100/−120 powder sizes.

Homogeneous and uniform distribution of wood flour and uniform wetting of polymer particles is necessary to improve the resulting composites’ properties.Reducing the aspect ratio increases the effect of filler reinforcement in composites and improves the stress distribution in the specimens.Finer particles than larger ones can create a more cohesive environment and better withstand the stresses applied.Impact energy increases with the increment of wood flour aspect ratio.

Additionally, the effect of injection mold temperature on the wood flour/polyethylene composite’s mechanical properties was studied. Three temperature levels of 170, 185, and 200 °C were investigated. As a result, it can be concluded that:Increasing the injection mold temperature up to 185 °C, better and sufficient heat transfer was performed, and the adhesion was improved. In addition, the tensile, flexural, and impact energy were increased.At 200 °C, the polymer’s adhesion properties, joint surface, and joint surface quality shrank, and the tensile and flexural strength decreased.As the injection mold temperature increased to 185 °C, the polymer’s stiffness increased, and the tensile and flexural modulus increased.The increase in flexural modulus indicates a decrease in the composite material deformation under load, which is a positive factor in engineering structures that must withstand a large load without deformation.

To complete our experiments and for more discussion on this topic, a rheological and structural analysis must be undertaken in subsequent experimental works.

## Figures and Tables

**Figure 1 materials-14-03406-f001:**
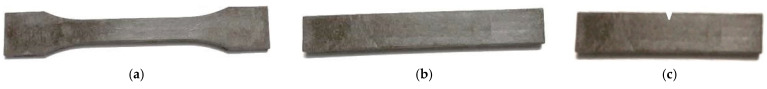
Samples for: (**a**) tensile; (**b**) bending; (**c**) impact test.

**Figure 2 materials-14-03406-f002:**
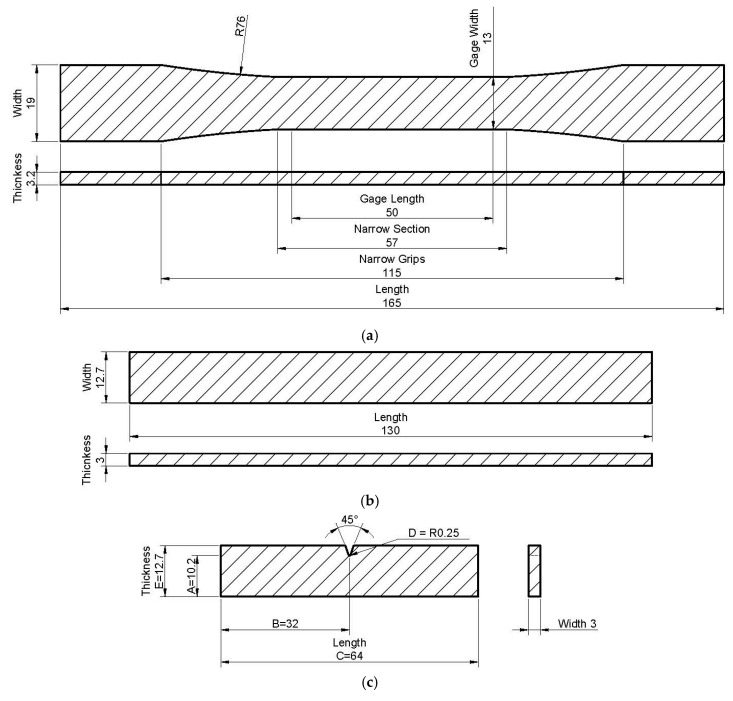
Dimension of specimens [mm] for (**a**) tensile; (**b**) bending; (**c**) impact test.

**Figure 3 materials-14-03406-f003:**
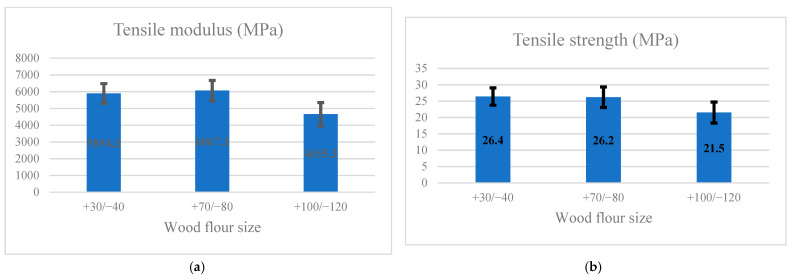
Impact of wood flour size on tensile (**a**) modulus; (**b**) strength.

**Figure 4 materials-14-03406-f004:**
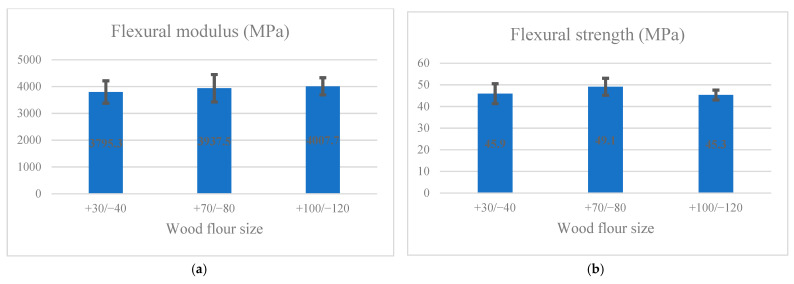
Impact of wood flour size on flexural (**a**) modulus; (**b**) strength.

**Figure 5 materials-14-03406-f005:**
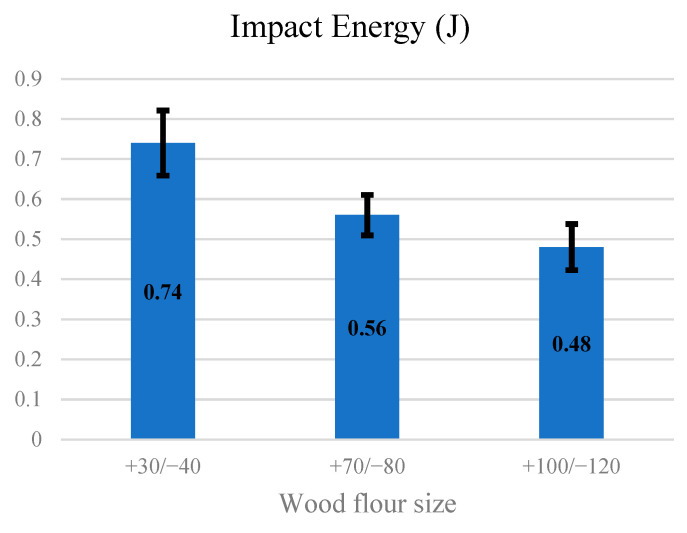
Effect of wood powder size on impact energy.

**Figure 6 materials-14-03406-f006:**
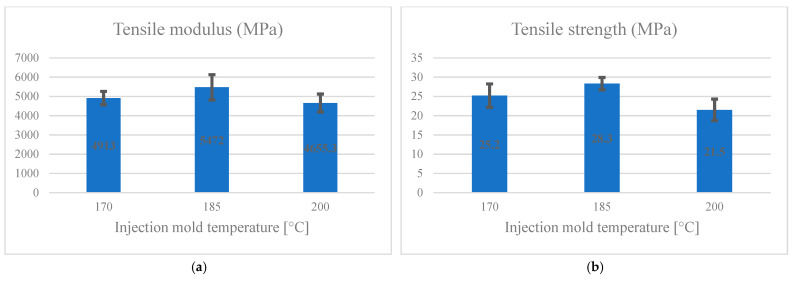
The effect of injection mold temperature on tensile (**a**) modulus; (**b**) strength.

**Figure 7 materials-14-03406-f007:**
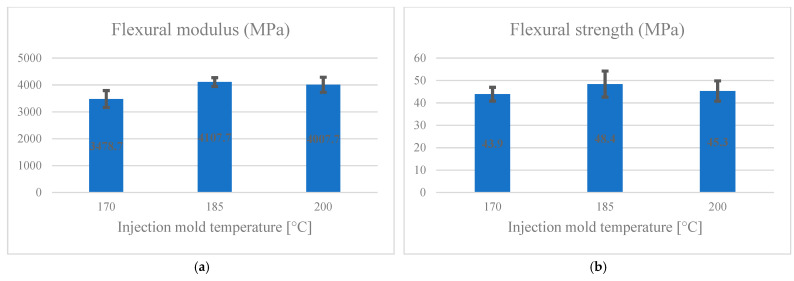
The effect of injection mold temperature on flexural (**a**) modulus; (**b**) strength.

**Figure 8 materials-14-03406-f008:**
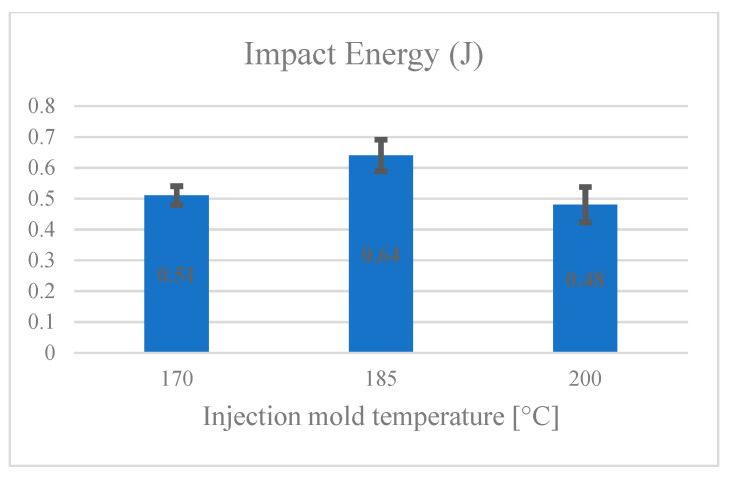
Effect of injection mold temperature on impact energy.

**Table 1 materials-14-03406-t001:** The percentage of composite components and injection mold temperature for specimens.

Specimen	MAPE ^1^ (%)	Heavy Polyethylene (%)	Wood Flour (%)	Mesh Dimensions	Injection Mold Temperature (°C)
1	3	50	50	+30/−40	200
2	3	50	50	+70/−80	200
3	3	50	50	+100/−120	200
4	3	50	50	+100/−120	170
5	3	50	50	+100/−120	185

^1^ Based on the total weight of the composite.

**Table 2 materials-14-03406-t002:** Mean length, diameter, and aspect ratio of wood flour dimensions.

Mesh Dimensions	Length (µm)	Diameter (µm)	Aspect Ratio (l/d)	Specimen
+30/−40	3239.99	1060.01	3.07	1
+70/−80	646.13	208.1	3.16	2
+100/−120	243.4	158.5	1.6	3, 4, 5

**Table 3 materials-14-03406-t003:** Variance analysis of wood flour size effect on the mechanical properties (*p*-values).

Mechanical Properties		*p*-Value
Tensile (MPa)	Strength	0.000 **
	Modulus	0.118
Flexural (MPa)	Strength	0.012 *
	Modulus	0.281
Impact energy (J)		0.003 *

* *p* < 0.05; ** *p* < 0.001.

**Table 4 materials-14-03406-t004:** Variance analysis of injection mold temperature effect on the mechanical properties (*p*-values).

Mechanical Properties		*p*-Value
Tensile (MPa)	Strength	0.000 **
	Moduli	0.631
Flexural (MPa)	Strength	0.222
	Moduli	0.124
Impact energy (J)		0.030 *

* *p* < 0.05; ** *p* < 0.001.

## Data Availability

Data available on request due to restrictions (privacy or ethical). The data presented in this study are available on request from the corresponding author. The data are not publicly available due to the privacy of results belonging to Islamic Azad University.

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
