# Peer review of "Investigation of Wood Flour Size, Aspect Ratios, and Injection Molding Temperature on Mechanical Properties of Wood Flour/Polyethylene Composites"

_materials, 2021, doi:10.3390/ma14123406_

Round 1
Reviewer 1 Report
The present manuscript, in my opinion, does not reach the level of the journal MATERIALS, which has a high impact factor. It is more like a description of a students' laboratory work with a small amount and low quality of their own obtained results and a large number of references to the work of other authors.
The abstract and conclusions require substantial revision. In particular, the abstract contains standards for measuring various characteristics, which is clearly superfluous. Also, the abstract and conclusions are just a short listing of the results obtained by the authors without any explanation.
The text of the manuscript contains sentences that are too long, confusing and very difficult to understand, such as on Page 3, Lines 98-104. There are also a large number of minor typos (lack of necessary spaces, non-use of superscripts in units of measurement, etc.).
The description of the preparation of samples is not complete enough: it is not clear how the mixture of wood flour with polyethylene was prepared, with the help of what equipment. Were the composite pellets from wood flour with PE preliminarily obtained, followed by injection molding of samples from these pellets, or was the dry mixture immediately loaded into the injection molding machine? What was the temperature in the molding form during injection molding?
The authors also mention the use of sieves with mesh sizes of 30, 70 and 100 mesh, but the results obtained are not then described for these sieves.
It is not clear what the error was when measuring the length and diameter of wood flour particles - there are no microscopic data.
Tables 3 and 4 are not clear and require additional explanation.
Measurement errors (bars) are not indicated anywhere on the strength and modulus graphs.
Periodically, when describing the results obtained, the authors mention a certain change in viscosity, although they did not conduct any rheological studies, although these studies would be very useful and informative for this work.
In addition, the authors try to speculate about the homogeneity of mixing wood flour with PE and about the improvement of the distribution of particles in the polymer matrix, as well as about adhesion between them. However, such discussions require experimental confirmation, for example, using a scanning electron microscope.
On Page 9, Lines 278-288, the authors discuss the possible degradation of the filler, although again they have no experimental results for this (e.g., TGA or IR spectroscopy).
Thus, this work contains practically no experimental data for a full description of the results obtained due to the absence of additional experimental research methods.
I would recommend that the authors conduct additional investigations (rheological, structural, etc.) in order to be able to describe the obtained mechanical properties of the studied samples on their basis, and not referring to other people's results in the literature.
Author Response
Dear Reviewer,
Please see the attachment.
Kind regards,
The Authors

Reviewer 2 Report
This paper investigates the processing conditions as well as filler type on the mechanical properties of wood flour/polyethylene composites.
I would recommend first some important modifications to the manuscript before considering it for publication in the Materials journal.
1) page 1, line 21: correct the sentence: "As the flour increased....". Quantity increased or mesh increased?
2) section 2.1: page 3, line 99. Such long statements must be made concise and easy to understand for the reader to gain interest. Kindly improve it.
3) section 2.4, page 3, line 131. It is not necessary to state the location of the laboratory location in the manuscript. Such can be done in Acknowledgment section only. Kindly remove. Also there are no images etc. showed from this analysis. Any reason still to state that? Insert images where necessary.
4) Table 2, page 4: Appearance coefficient does not seems to be a scientific term here. I think Aspect ratio should be used as the term appropriate here.
5) Line 144, page 4: Apparent coefficient or appearance coefficient? Are those the same? replace such term(s) with aspect ratio as mentioned before.
6) It is highly recommended to insert error bars as standard deviation in the Figures 1 to 6.
7) Line 210, page 6: I think the author intended to say that with filler dimensions of 40 mesh, impact strength was high whereas it was low in case of 120 mesh? Figure 3 says the opposite of what the line 210 states. Kindly check.
Author Response

(The authors gave the same response as above.)

Reviewer 3 Report
I read with interest the article "Investigation of Wood Flour Size, Aspect Ratios, and Injection Moulding Temperature on Mechanical Properties of Wood Flour/Polyethylene Composites". In this work, the authors investigated the effects of wood flour size, aspect ratios, and injection molding temperature on the mechanical properties of wood flour/polyethylene composites. The wood flour was at a constant 50%, and the flour size at 40, 80 and 120 mesh. There are several works that address the properties of HDPE and wood flour. The originality of the present work is mainly related to the effect of reinforcement size on the mechanical properties of the wood-plastic composite. This manuscript is composed of 12 pages including 6 figures and 38 references. The latter all reflect the subject matter addressed here.
Thus, the review of the article leads to the following observations: The introduction should be enriched mainly by recent and complementary work done on HDPE composites and wood flour such as:
- The impact of pathogens and fungi on the mechanical properties of the composite. In this regard, I suggest the following works: Tazi M., Erchiqui, Kaddami H. (2018). Influence of softwood filler content on biodegradability and morphological properties of wood-polyethylene composites. Polymer Composites. 39 : 29-37
- - Effect of wood flour mass concentration on the rheological properties of wood flour-based HDPE composites: Godard, F., M. Vincent, J.-F. Agassant, and B. Vergnes. Rheological behaviour and mechanical properties of sawdust/polyethylene composite, Journal of Applied Polymer Science, vol. 112, pp. 2559-2566, 2009.
2. Authors should add data on heat capacity and specific volume as a function of filler size and temperature.
3. If possible, the authors should add thermal stability curves for wood flour and HDPE composites as a function of reinforcement size.
4. I encourage the authors to read the paper carefully to publish the typos ;
5. Finally, I encourage authors to submit their paper according to the journal's instructions.
Based on what has been said, this manuscript may eventually be published in the journal, if the authors are willing to carefully execute the above comments and questions. In addition, I agree to revise this article if necessary.
Author Response

(The authors gave the same response as above.)

Round 2
Reviewer 1 Report
Please, see the attached file!

Author Response
Dear Reviewer,
Please see the attachment.
Kind regards,
Authors

Reviewer 2 Report
The manuscript has been corrected as per the referees comments. It can be now accepted for publication.
Author Response
Dear Reviewer,
The authors would like to express their great appreciation for your comment and review of the manuscript once again.
The authors thank the Reviewer for his final decision and all contribution.
Kind regards,
Authors
Reviewer 3 Report
The new updated version of the article titled "Investigation of Wood Flour Size, Aspect Ratios, and Injection Moulding Temperature on Mechanical Properties of Wood Flour/Polyethylene Composites" is consistent with my remarks and deserves to be published in the journal Materials
Author Response

(The authors gave the same response as above.)
